# A tonic nicotinic brake controls spike timing in striatal spiny projection neurons

Lior Matityahu[1†], Jeffrey M Malgady[2†], Meital Schirelman[1†], Yvonne Johansson[3‡], Jennifer A Wilking[2], Gilad Silberberg[3], Joshua A Goldberg[1*], Joshua L Plotkin[2*]

[1]Department of Medical Neurobiology, Institute of Medical Research Israel–Canada, The Faculty of Medicine, The Hebrew University of Jerusalem, Jerusalem, Israel; [2]Department of Neurobiology and Behavior, Center for Nervous System Disorders, Stony Brook University School of Medicine, Stony Brook University, Stony Brook, United States; [3]Department of Neuroscience, Karolinska Institutet, Stockholm, Sweden

**\*For correspondence:**
joshua.goldberg2@mail.huji.ac.il (JAG);
joshua.plotkin@stonybrook.edu (JLP)

[†]These authors contributed equally to this work

**Present address:** [‡]Sainsbury Wellcome Centre for Neural Circuits and Behaviour, University College London, London, United Kingdom

**Abstract** Striatal spiny projection neurons (SPNs) transform convergent excitatory corticostriatal inputs into an inhibitory signal that shapes basal ganglia output. This process is fine-tuned by striatal GABAergic interneurons (GINs), which receive overlapping cortical inputs and mediate rapid corticostriatal feedforward inhibition of SPNs. Adding another level of control, cholinergic interneurons (CINs), which are also vigorously activated by corticostriatal excitation, can disynaptically inhibit SPNs by activating α4β2 nicotinic acetylcholine receptors (nAChRs) on various GINs. Measurements of this disynaptic inhibitory pathway, however, indicate that it is too slow to compete with direct GIN-mediated feedforward inhibition. Moreover, functional nAChRs are also present on populations of GINs that respond only weakly to phasic activation of CINs, such as parvalbumin-positive fast-spiking interneurons (PV-FSIs), making the overall role of nAChRs in shaping striatal synaptic integration unclear. Using acute striatal slices from mice we show that upon synchronous optogenetic activation of corticostriatal projections blockade of α4β2 nAChRs shortened SPN spike latencies and increased postsynaptic depolarizations. The nAChR-dependent inhibition was mediated by downstream GABA release, and data suggest that the GABA source was not limited to GINs that respond strongly to phasic CIN activation. In particular, the observed decrease in spike latency caused by nAChR blockade was associated with a diminished frequency of spontaneous inhibitory postsynaptic currents in SPNs, a parallel hyperpolarization of PV-FSIs, and was occluded by pharmacologically preventing cortical activation of PV-FSIs. Taken together, we describe a role for tonic (as opposed to phasic) activation of nAChRs in striatal function. We conclude that tonic activation of nAChRs by CINs maintains a GABAergic brake on cortically-driven striatal output by 'priming' feedforward inhibition, a process that may shape SPN spike timing, striatal processing, and synaptic plasticity.

## Editor's evaluation

Matityahu et al., investigate the influence of nicotinic acetylcholine receptor signaling on striatal microcircuit function through a combination of slice electrophysiology, optogenetics, and pharmacology. They find that nicotinic signaling delays spiking of striatal projection neurons in response to excitatory input, likely through the tonic release of acetylcholine by cholinergic interneurons onto local GABAergic interneurons and their influence on striatal projection neurons. Understanding how acetylcholine shapes striatal circuits is important, as this neurotransmitter is implicated in multiple movement disorders as well as other basal ganglia-related diseases.

## Introduction

As the main input nucleus of the basal ganglia, the striatum receives convergent cortical synaptic information, generating a computationally transformed interpretation of this information and relaying it to intermediate and output nuclei of the basal ganglia. Striatal output is carried exclusively by GABAergic striatal spiny projection neurons (SPNs), which account for ~95% of all striatal neurons (*Gerfen and Surmeier, 2011*; *Silberberg and Bolam, 2015*). The remaining 5% of striatal neurons are interneurons, most of which are GABAergic interneurons (GINs). There are several molecularly distinct classes of GINs that exhibit diverse intrinsic firing patterns and patterns of intrastriatal connectivity (*Tepper et al., 2010*; *Muñoz-Manchado et al., 2018*; *Assous and Tepper, 2019*). In the context of feedforward cortico- and thalamo-striatal transmission, the GINs' function lies in the feedforward inhibition they provide to SPNs (*Tepper et al., 2004b*; *Assous et al., 2017*). Such inhibition can fashion striatal output by perturbing the precise timing of SPNs or prevent their spiking altogether. The only non-GINs are the large, aspiny, cholinergic interneurons (CINs) (*DiFiglia, 1987*; *Tepper and Bolam, 2004a*, *Plotkin and Goldberg, 2019*; *Poppi et al., 2021*). Despite their small numbers (approximately 1% of striatal neurons), the acetylcholine (ACh) they release influences the entire striatal microcircuitry. A single CIN axon can cover a third of the striatum (in linear dimension) and has ACh release sites every few micrometers (*DiFiglia, 1987*). In fact, the striatum has the highest expression of cholinergic markers in the entire CNS (*Mesulam et al., 1992*; *Contant et al., 1996*). CINs are autonomous pacemakers (*Bennett and Wilson, 1999*) that are largely identified as the tonically active neurons (TANs) of the striatum (*Kimura et al., 1984*; *Wilson et al., 1990*; *Aosaki et al., 1994*; *Aosaki et al., 1995*; *Morris et al., 2004*), firing 3–10 spikes/s in vivo. On the backdrop of this ongoing discharge, the main signal they convey in vivo is a pause in response to primary reward or salient stimuli associated with reward (*Aosaki et al., 1994*; *Morris et al., 2004*; *Goldberg and Reynolds, 2011*; *Apicella, 2017*). The duration of the pause in response to a brief conditioned sensory stimulus is on the order of 200–300 ms (*Kimura et al., 1984*; *Raz et al., 1996*; *Apicella et al., 1997*).

CINs regulate SPNs in three main ways. The best characterized regulation is exerted via muscarinic ACh receptors (mAChRs). Activation of presynaptic and postsynaptic mAChRs on SPNs modulates synaptic transmission, synaptic plasticity, and the intrinsic excitability of SPNs by modulating various voltage-activated $Ca^{2+}$ and $K^+$ channels (*Akins et al., 1990*; *Calabresi et al., 1999*; *Gabel and Nisenbaum, 1999*; *Day et al., 2008*; *Goldberg et al., 2012*; *Zucca et al., 2018*). CINs influence SPNs via nicotinic ACh receptors (nAChRs) as well, albeit indirectly. Activation of nAChRs on striatal dopaminergic fibers can evoke dopamine (DA) release (*Zhou et al., 2001*; *Cachope et al., 2012*; *Threlfell et al., 2012*; *Liu et al., 2022*) which, in turn, can modulate the intrinsic excitability of SPNs and synaptic transmission and plasticity at synaptic inputs. All of the above influences involve GPCR-linked ACh and DA receptors, which means that they are unlikely to dynamically affect the moment-by-moment processing and transmission of excitatory inputs to SPNs. Indeed, a recent study in which CINs were silenced optogenetically in vivo has put a lower bound on how fast mAChR-mediated effects can come into play. When CINs are synchronously silenced for over 500 ms, SPNs begin to show signs of mAChR-dependent reductions in excitability. However, that effect comes into play only after a 400 ms delay (*Zucca et al., 2018*). The only known mechanisms by which CINs can rapidly affect SPN activity (sooner than 400 ms) involve nicotinic or muscarinic modulation of synaptic glutamate release or α4β2 nAChR-dependent GABA release onto the SPNs' ionotropic GABA$_A$ receptors (*Pakhotin and Bracci, 2007*; *English et al., 2011*; *Faust et al., 2015*; *Faust et al., 2016*; *Assous et al., 2017*; *Assous, 2021*). The α4β2 nAChRs that drive disynaptic GABA release are located on GINs (*English et al., 2011*; *Faust et al., 2015*), and on dopaminergic nigrostriatal terminals (*Nelson et al., 2014b*).

Because CINs receive massive excitatory cortical and thalamic input (*Lapper and Bolam, 1992*; *Thomas et al., 2000*; *Mamaligas et al., 2019*), it is conceivable that their disynaptic inhibition of SPNs comes into play during bouts of cortical or thalamic activity. However, this raises a conceptual problem. Because cortex and thalamus can activate feedforward inhibition to SPNs via various striatal GINs (e.g., cortex → GIN → SPN), what is gained by recruiting an additional pathway that is necessarily slower and less reliable due to its additional synapse (e.g., cortex → CIN → GIN → SPN)? Moreover, what is the role of nAChRs on GINs, such as the parvalbumin-positive fast-spiking interneurons (PV-FSIs) that only weakly respond to direct activation of CINs (*English et al., 2011*; *Nelson et al., 2014a*; *Nelson et al., 2014b*)? This conceptual problem exists only if we assume that the role of CINs in modulating SPN function through the activation of nAChRs is anchored around their phasic activation

by excitatory afferents. In this study we demonstrate that ongoing CIN activity exerts a constant brake on SPN excitation and spike initiation via tonically activated α4β2 nAChRs located on discrete populations of GINs. We uncover these nicotinic effects using transgenic mice that nominally express channelrhodopsin-2 (ChR2) in corticostriatal fibers (*Arenkiel et al., 2007*). Activation of these fibers creates a competition between monosynaptic cortical excitation and polysynaptic feedforward inhibition to SPNs. We show that nAChRs embedded in a GABAergic postsynaptic pathway are capable of controlling SPN spike timing with millisecond precision.

## Results

To interrogate microcircuit-level responses of SPNs to cortical excitation of the striatum, we generated acute ex vivo brain slices from Thy1-ChR2 mice, which nominally express ChR2 in cortical neurons

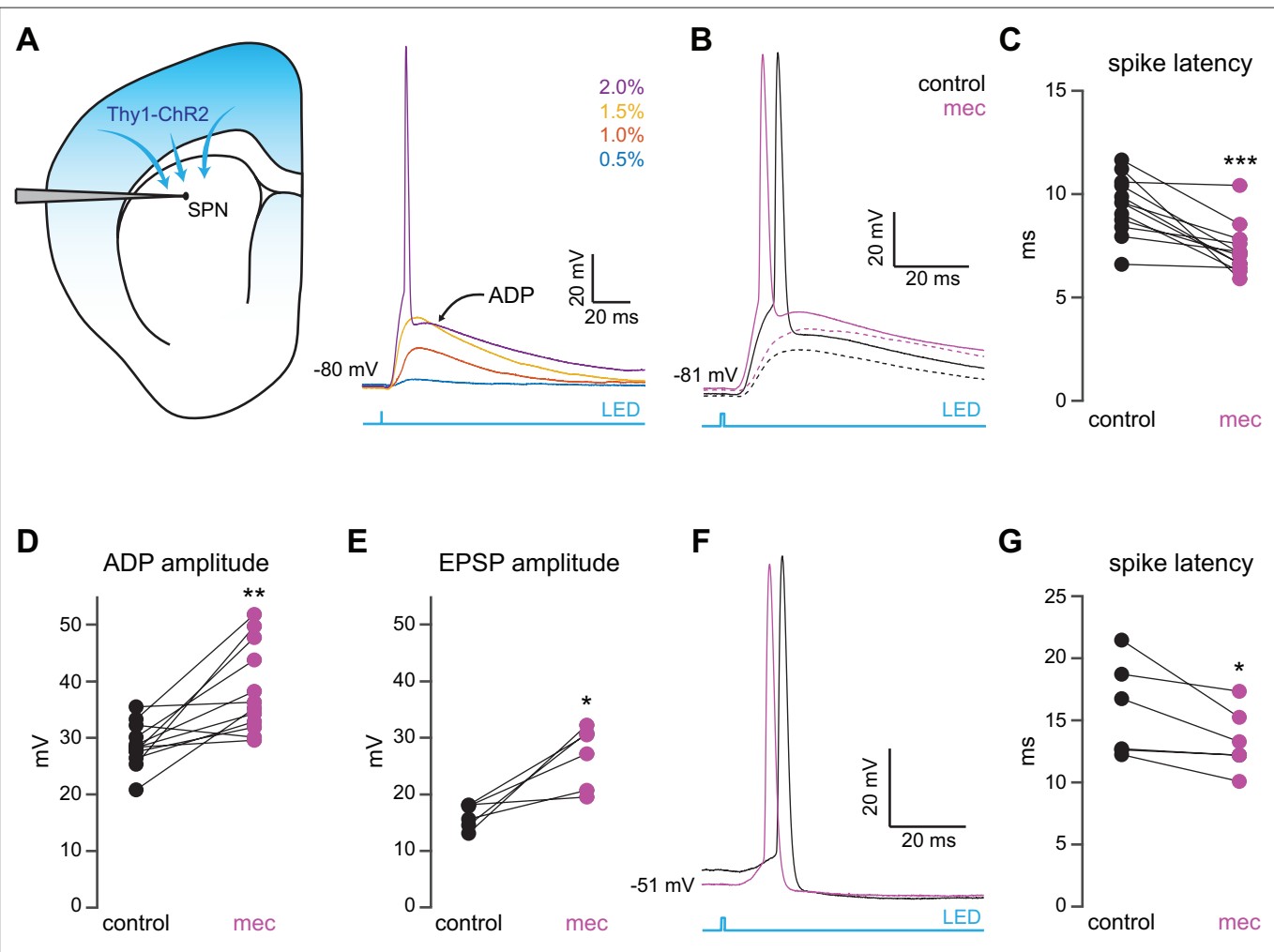

**Figure 1.** Nicotinic acetylcholine receptor (nAChR)-dependent inhibition and delay of spike latency in spiny projection neurons (SPNs) activated by corticostriatal fibers. (**A**) Left: diagram of recording configuration. An SPN is patched in an acute slice from a Thy1-ChR2 mouse. Right: 1-ms-long 470 nm LED pulse of increasing intensity generates excitatory postsynaptic potentials (EPSPs), or an AP followed by an afterdepolarization (ADP) in an SPN. (**B**) Examples of the effect of 10 μM mecamylamine (mec), an nAChR antagonist, on EPSP amplitude (dashed) or spike latency and ADP amplitude (solid). (**C–E**) Mecamylamine significantly shortens spike latencies (p=4.8·10⁻⁴, n=12 SPNs, signed-rank test [SRT]) (**C**), as well as ADP (p=2.4·10⁻³, n=12 SPNs, SRT) (**D**) and EPSP (p=0.03, n=6 SPNs, SRT) (**E**) amplitudes. (**F**) Examples of the effect of mecamylamine on latency of a spike triggered synaptically from a depolarized potential with an LED pulse. (**G**) Mecamylamine significantly shortens spike latencies in SPNs held at a depolarized potential (p=0.03, n=6 SPNs, SRT). Two-sided Wilcoxon SRT. \*\*\*p<0.001, \*\*p<0.01, \*p<0.05.

The online version of this article includes the following figure supplement(s) for figure 1:

**Figure supplement 1.** Dihydro-β-erythroidine hydrobromide (DHβE) mimics the effect of mecamylamine on striatal spiny projection neuron (SPN) spike latency and afterdepolarization (ADP) amplitude.

(*Arenkiel et al., 2007*; *Aceves Buendia et al., 2019*). Full-field optogenetic stimulation with a 470 nm LED engaged corticostriatal afferents and generated in SPNs either: excitatory postsynaptic potentials (EPSPs) for low LED intensities, or – in the suprathreshold condition – an action potential (AP) followed by an afterdepolarization (ADP) lasting 10 s of milliseconds (*Figure 1a*). While ADPs were kinetically similar to EPSPs under our experimental conditions, we explicitly separated these measures because ADPs (1) can be modulated by similar cortically activated local striatal circuits that shape SPN spike timing and (2) peak closer to spike threshold, positioning them to influence subsequent spike generation (*Flores-Barrera et al., 2010*). Blockade of nAChRs with mecamylamine (mec; 10 μM) enhanced SPN responses to cortical stimulation (*Figure 1b*), as it shortened AP latency (*Figure 1c*) and increased both the ADP (*Figure 1d*) and subthreshold EPSP amplitudes (*Figure 1e*). Because this suggests that under these conditions EPSPs and ADPs are mechanistically similar, the remainder of this study focuses on SPN AP latency and ADP amplitude generated by just-suprathreshold LED stimulation, which facilitated comparison among cells. The effects of mecamylamine were replicated in the presence of the α4β2 nAChR-selective antagonist, dihydro-β-erythroidine hydrobromide (DHβE) (10 μM, AP latency: n=8 SPNs, p=7.8·10⁻³, signed-rank test (SRT); ADP amplitude: n=7 SPNs, p=0.03, SRT; *Figure 1—figure supplement 1*). Because the resting membrane potential of SPNs is very hyperpolarized compared to the depolarized potential from which spikes typically occur in vivo (*Wilson and Groves, 1981*; *Stern et al., 1998*), we repeated the experiment while holding the SPN with a constant positive current injection in the just-subthreshold region (e.g., in the –55 to –50 mV range). Mecamylamine shortened the latency to AP from this depolarized potential as well (*Figure 1f–g*). We note that ADPs were not consistently present under these conditions, likely because the depolarized potential typically eclipsed ADP amplitudes observed from rest.

Because SPNs themselves do not express nAChRs, the action of mecamylamine must be indirect. Functional nAChRs are expressed at multiple loci of the striatal circuit, including afferent axonal terminals and intrastriatal interneurons (*Nelson et al., 2014b*; *Faust et al., 2016*; *Abudukeyoumu et al., 2019*; *Assous, 2021*; *Abbondanza et al., 2022*; *Morgenstern et al., 2022*). While the broad antagonistic actions of mecamylamine and DHβE should block nicotinic receptors at most of these loci, α7 nAChRs, which are predominantly expressed at corticostriatal axon terminals, may be spared (*Solinas et al., 2007*; *Licheri et al., 2018*; *Assous, 2021*). Indeed, mecamylamine did not have a significant effect on the strength or release probability of corticostriatal glutamatergic afferents, as measured by local electrical stimulation (*Figure 2A-C*). Blockade of α7 nAChRs with the more selective α7 antagonist methyllycaconitine (MLA; 5 μM) did not significantly alter corticostriatal afferent strength or release probability either (*Figure 2—figure supplement 1*), suggesting that nAChRs on these synapses are not poised to track tonic ACh release.

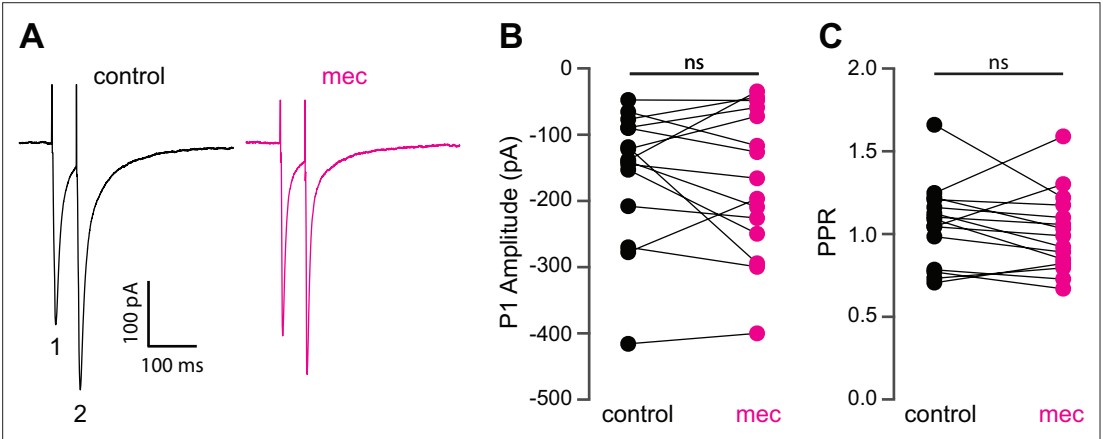

**Figure 2.** Mecamylamine does not alter release probability of excitatory synaptic inputs. (**A**) Example responses of an striatal spiny projection neuron (SPN) to paired pulse stimulation (evoked by two local electrical stimuli separated by 50 ms), before and in the presence of mecamylamine (10 μM). (**B**) Mecamylamine had no effect on the excitatory postsynaptic current (EPSC) amplitude evoked by the first paired stimulus (**P1**) (p=0.59, n=15 SPNs, signed-rank test [SRT]). (**C**) Mecamylamine had no effect on the paired pulse ratio (PPR) (P2/P1, p=0.33, n=15 SPNs, SRT).

The online version of this article includes the following figure supplement(s) for figure 2:

**Figure supplement 1.** Methyllycaconitine (MLA) does not alter release probablity of excitatory synaptic inputs.

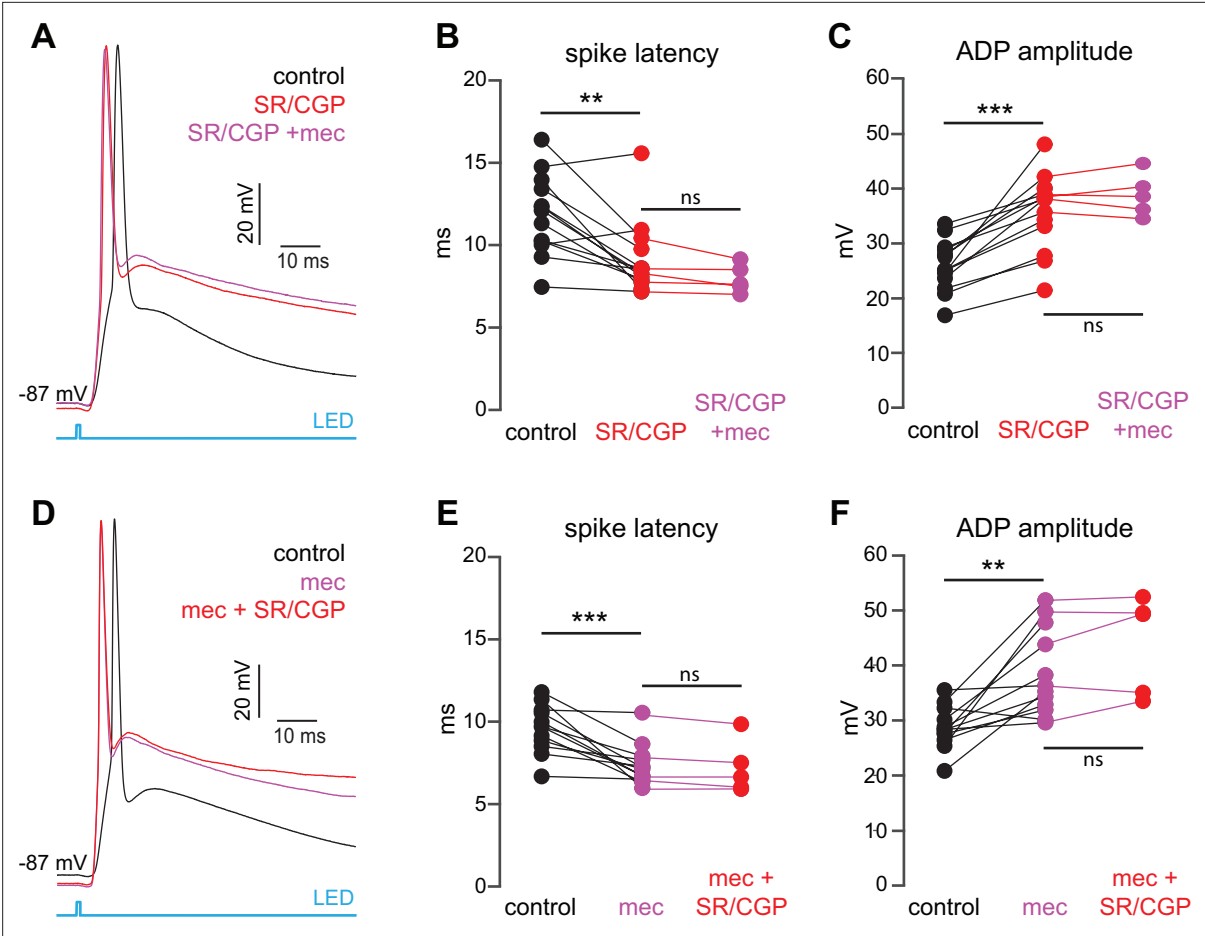

**Figure 3.** The nicotinic acetylcholine receptor (nAChR)-dependent inhibition of corticostriatal striatal spiny projection neuron (SPN) activation is mediated through and saturates GABAergic inhibition. (**A**) Example of the occlusion of the mecamylamine effect on optogenetic synaptic activation of SPNs by receptor (GABAR) antagonists, 10 μM SR-95531 (GABA$_A$R antagonist) and 2 μM CGP-55845 (GABA$_B$R antagonist). Distribution of spike latencies (**B**) and ADP amplitude (**C**) in response to application of GABAR antagonists followed by mecamylamine, showing that application of GABAR antagonist significantly shortens the action potential (AP) latency (p=4.6·10$^{-3}$, n=13 SPNs, signed-rank test [SRT]) and enhancement of the ADP amplitude (p=2.4·10$^{-4}$, n=13 SPNs, SRT). In contrast, the subsequent mecamylamine application fails to further shorten the AP latency (p=0.0625, n=5 SPNs, SRT) or further enhance the ADP amplitude (p=1, n=5, SRT). (**D**) Example of how the mecamylamine effect saturates the GABAergic inhibition of the optogenetic synaptic activation of SPNs. (**E–F**) Distribution of spike latencies (**E**) and ADP amplitude (**F**) in response to application of mecamylamine followed by GABAR antagonists, showing that the subsequent application of GABAR antagonists fails to further shorten the AP latency (p=0.25, n=5 SPNs, SRT) or further enhance the ADP amplitude (p=0.44, n=5 SPNs, SRT).

The online version of this article includes the following figure supplement(s) for figure 3:

**Figure supplement 1.** ChR2 expression is not observed in cortical or striatal GABAergic neurons in Thy1-ChR2 mice.

The lack of an effect of mecamylamine on presynaptic release probability suggests that mecamylamine shapes SPN spike timing through postsynaptically expressed nicotinic receptors within the striatum. CINs form a disynaptic circuit with SPNs, which is mediated by neuropeptide Y-neurogliaform (NPY-NGF) GINs (and perhaps other classes of nAChR-expressing GINs) in an nAChR-dependent manner (*English et al., 2011*; *Elghaba et al., 2016*; *Faust et al., 2016*; *Assous et al., 2017*; *Tepper et al., 2018*). Consistent with the involvement of such an inhibitory polysynaptic circuit, we note that blockade of nAChRs reduced the latency of synaptically evoked spikes in SPNs that were held just-subthreshold even if the SPN membrane potential fell (*Figure 1f*), an observation that could be accounted for by GABA receptor-mediated shunting (*Gustafson et al., 2006*). Accordingly, blockade of GABAergic transmission with the GABA$_A$ and GABA$_B$ receptor antagonists SR-95531 (10 μM) and CGP-55845 (2 μM), respectively, mimicked the effect of mecamylamine on spike timing and ADP amplitude (*Figure 3a–c*). Blockade of GABA receptors fully occluded mecamylamine's effect on both

spike latency and ADP amplitude (*Figure 3a–c*), further implicating a CIN-GIN-SPN disynaptic circuit in mediating the actions of nAChRs. Strikingly, mecamylamine prevented GABA receptor antagonists from further advancing spike timing or enhancing ADP amplitude, suggesting that nAChR activation can saturate GABAergic inhibition (*Figure 3d–f*).

While SPN spiking was driven by convergently activated corticostriatal inputs in the above experiments, activation of monosynaptic GABAergic inputs due to ectopic expression of ChR2 cannot necessarily be ruled out. This is an important point, because mecamylamine's mechanism of action involves nAChRs embedded in a GABAergic circuit. Immunohistochemical staining did not show any evidence of ChR2 expression in GABAergic neurons within the cortex or striatum, including somatostatin (SOM)-expressing cortical neurons known to directly innervate SPNs (*Rock et al., 2016*), but did reveal ChR2-positive neurons in the globus pallidus (*Figure 3—figure supplement 1*). This is in addition to observations of ChR2 expression in GABAergic neurons of the substantia nigra pars reticulata in Thy1-ChR2 mice (*Pan et al., 2013*; *Higgs and Wilson, 2017*; *Tiroshi and Goldberg, 2019*). Indeed, blockade of glutamate receptors to eliminate feedforward inhibition revealed the presence of an optogenetically evoked monosynaptic GABAergic input to SPNs (*Figure 3—figure supplement 1*). This monosynaptic input was insensitive to mecamylamine, however, making it unlikely to mediate the observed effect of nAChRs on SPN spike timing (*Figure 3—figure supplement 1*).

CINs receive converging excitatory inputs from the thalamus as well as the cortex (*Lapper and Bolam, 1992*; *Thomas et al., 2000*). Indeed, targeted recordings from CINs confirmed that they are robustly engaged by our corticostriatal stimulation protocol, even displaying a lower stimulation threshold and shorter average delay to spike than SPNs (*Figure 4a–c*). Despite the observed high fidelity and speed of cortically evoked CIN spiking, a comparison of the temporal dynamics of synchronous cortical engagement of SPN inhibition (latency of approximately 5 ms, *Figure 5a*) vs. synchronous CIN engagement of GINs (latency of approximately 11 ms, *English et al., 2011*; *Nelson et al., 2014b*) shows that phasic activation of CINs is still not sufficiently fast to account for the mecamylamine-induced spike delay we observed in SPNs (*Figure 5b*). In particular, even if there are sufficient numbers of CINs that respond *instantaneously* to cortical activation

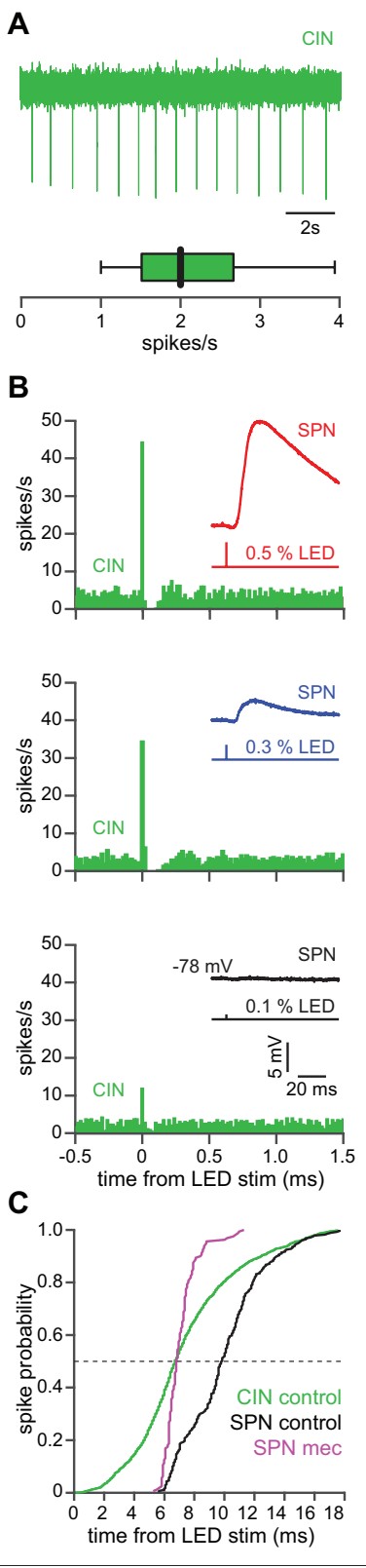

**Figure 4.** The autonomous firing of cholinergic interneurons (CINs) enables them to respond more rapidly and vigorously than striatal spiny projection neurons (SPNs) to feedforward corticostriatal excitation.

*Figure 4 continued on next page*

*Figure 4 continued*

(**A**) Example of the autonomous discharge of a CIN recorded in cell-attached configuration (top) and the distribution of firing rates (bottom, n=15 CINs) in an acute striatal slice. (**B**) Peristimulus time histograms (PSTHs) of a CIN in response to optogenetic activation of corticostriatal fibers at various 470 nm LED intensities as compared to the amplitude of the excitatory postsynaptic potentials (EPSPs) evoked in a nearby SPN (insets), demonstrating that CINs are much more sensitive than SPNs to cortical activation. (**C**) Cumulative distribution of the latency to first spike of CINs recorded in cell-attached mode (green, n=8 CINs) as compared to the latency to first spike in an SPN recorded (from a resting state) in the whole-cell mode (n=12), before (black) and after (magenta) application of mecamylamine.

due to their ongoing activity, the earliest latency at which their phasic inhibition can influence SPNs is 11 ms later (*English et al., 2011*; *Nelson et al., 2014b*). However, SPN spikes are being delayed beginning 5 ms after stimulation (*Figure 5c* magenta), which cannot be explained by phasic CIN activation.

If phasic activation of nAChRs by synaptically evoked CIN activity cannot account for the observed delay in SPN spike initiation (due to the necessarily slow nature of the CIN-GIN-SPN disynaptic signal), how does blocking nAChRs advance synaptically evoked SPN spiking? Given the autonomous pacemaking nature of CINs, perhaps the explanation is that *tonic*, rather than phasic, activation of nAChRs is key to the phenomenon. There are three obvious mechanisms that could be at play in this scenario: (1) ongoing nAChR activation (either tonic or phasic activation that is not time-locked with the corticostriatal stimulation event leading to SPN spiking) may decrease SPN intrinsic excitability indirectly, likely by altering ongoing neuromodulator release (*Zhou et al., 2002*; *Rice and Cragg, 2004*); (2) the effect on SPN spiking may be a non-specific drug effect of mecamylamine (this is unlikely, since DHβE had the same effects); or (3) *tonic* activation of somatodendritic nAChRs may produce an ongoing depolarization of GINs, ultimately priming or accentuating corticostriatal feedforward inhibition.

To test the possibility that blockade of ongoing nAChR activation increases SPN excitability, we measured the current-voltage (IV) relationship of SPNs before and during mecamylamine application. While there is no indication that our detected effect of mecamylamine on spike timing or ADP amplitude is bimodal and specific to direct pathway SPNs (dSPNs) or indirect pathway SPNs (iSPNs), dSPNs and iSPNs display differences in basal excitability (*Gertler et al., 2008*), which can complicate interpretation. We therefore performed these experiments in transgenic mice where dSPNs and iSPNs could be identified by their fluorescent label (*Shuen et al., 2008*; *Ade et al., 2011*; *Figure 6a*).

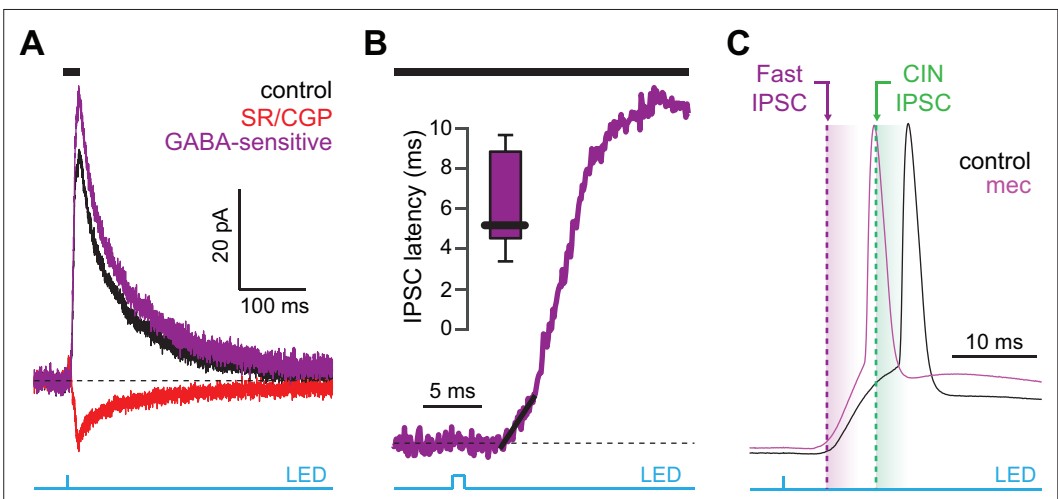

**Figure 5.** Latency of feedforward GABAergic inhibitory postsynaptic currents (IPSCs) in striatal spiny projection neurons (SPNs) activated synaptically by corticostriatal fibers. (**A**) Left: IPSCs recorded in an SPN held at +10 mV before (black) and after (red) application of GABAR antagonists, reveals GABA-sensitive (purple) IPSC. Right: zoom-in of the GABAergic IPSCs enables estimation (fit of diagonal black line) of the latency of the feedforward GABA-sensitive inhibition. (**B**) Distribution of the latency of feedforward GABAergic inhibition to SPNs in the Thy1-ChR2 mouse (n=7 SPNs). (**C**) Schematic demonstrating that the feedforward GABAergic inhibition precedes the earliest timing of feedforward cholinergic disynaptic inhibition of SPNs, indicating that phasic activation of CINs cannot explain the nicotinic acetylcholine receptor (nAChR)-dependent delay of spike latency.

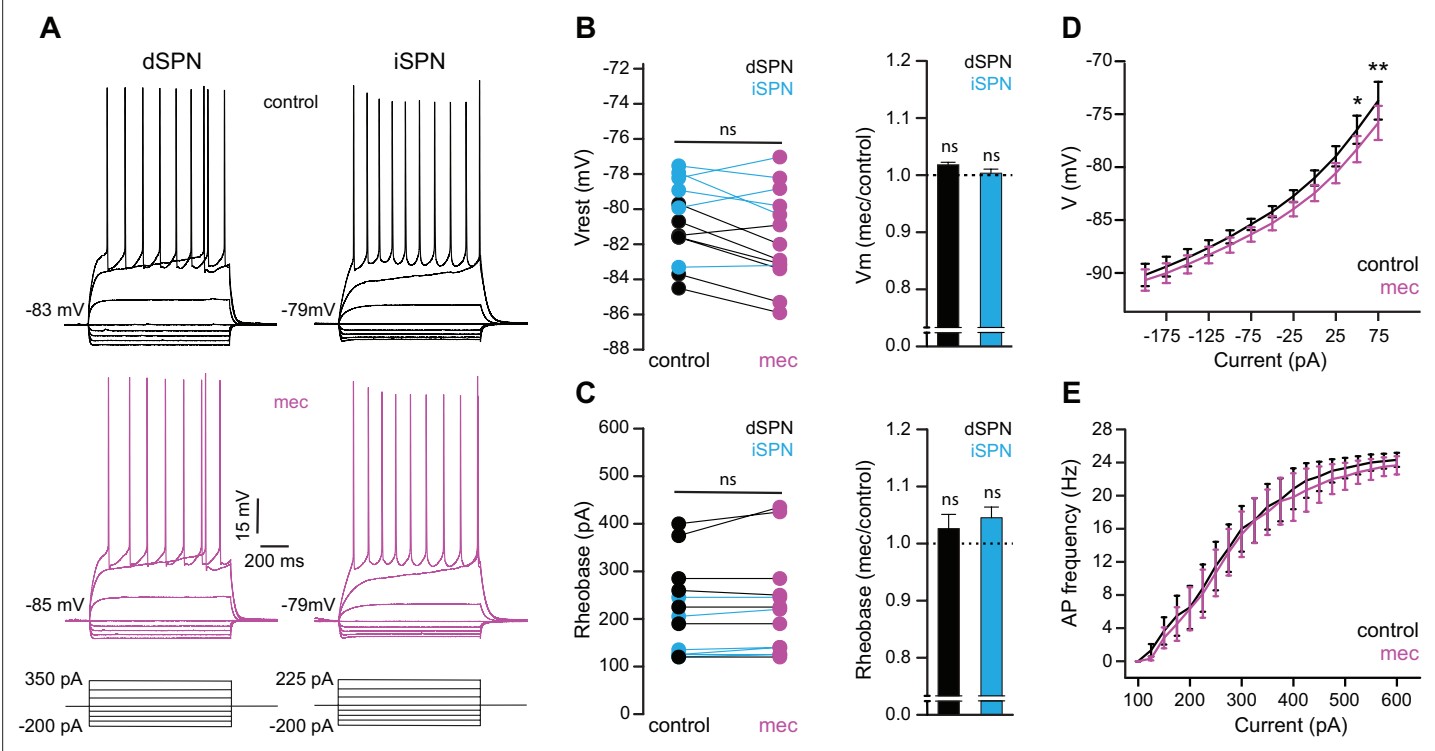

**Figure 6.** Striatal spiny projection neuron (SPN) intrinsic excitability is not increased by nicotinic acetylcholine receptor (nAChR) blockade. (**A**) Example current-voltage (IV) traces of direct pathway SPNs (dSPNs) and indirect pathway SPNs (iSPNs) in the absence and presence of 10 μM mecamylamine. (**B**) Left: resting membrane potential of dSPNs (black) and iSPNs (blue) in the absence and presence of mecamylamine (all SPNs: p=0.43, n=13; dSPNs: p=0.16, n=7; iSPNs: p=0.87, n=6; signed-rank test [SRT]). Right: percent change in resting membrane potential after mecamylamine application. (**C**) Left: rheobase current of dSPNs and iSPNs in the absence and presence of mecamylamine (all SPNs: p=0.075, n=13; dSPNs: p=0.5, n=7; iSPNs: p=0.095, n=6; SRT). Right: percent change in rheobase current after mecamylamine application. (**D**) Voltage responses to subthreshold current injections are not enhanced by mecamylamine (F(11,120)=0.86, p=0.58; n=11; two-way ANOVA), though a post hoc Bonferroni test reveals a significant mecamylamine-induced decrease at 50 pA (p<0.05) and 75 pA (p<0.01) current injections. (**E**) Action potential (AP) firing frequencies in response to suprathreshold somatic current injections were unaffected by mecamylamine (F(20,231)=0.43, p=0.99; n=12; two-way ANOVA).

The online version of this article includes the following figure supplement(s) for figure 6:

**Figure supplement 1.** Mecamylamine does not shorten the latency of striatal spiny projection neuron (SPN) action potentials (APs) induced by somatic current injection, or alter somatic excitability of SPNs from Thy1-ChR2 mice.

Mecamylamine had no significant effect on the resting membrane potential or rheobase current of either SPN type (*Figure 6b and c*). Furthermore, mecamylamine did not decrease the time to spike in response to somatic rheobase current injection or the latency to first spike in an AP train induced by suprathreshold somatic current injection (*Figure 6—figure supplement 1*). Given that most data to this point came from pooled populations of SPNs, and that we observed no SPN-type-specific effects of mecamylamine on SPN excitability, we performed additional analysis on pooled SPN data. While rheobase currents were unaffected, mecamylamine decreased the responsiveness to several sub-rheobase amplitude currents, arguing that if anything mecamylamine may *decrease* the excitability of SPNs in some regards (*Figure 6d*). Mecamylamine had no effect on the firing rate of SPNs (*Figure 6e*), nor did it alter IV properties of unidentified SPNs recorded from Thy1-ChR2 mice (*Figure 6—figure supplement 1*). Taken together, reduced spike latency induced by nAChR blockade cannot be explained by changes in SPN intrinsic excitability.

Because mecamylamine did not increase the intrinsic excitability of SPNs, we tested if nAChR blockade attenuated the basal inhibitory GABAergic influence that they are under. Indeed, mecamylamine significantly reduced the frequency of spontaneous inhibitory postsynaptic currents (sIPSCs) in a mixed population of SPNs (*Figure 7a–c*). As SPNs lack nAChRs, the source of these attenuated sIPSCs is likely GINs. Indeed, various classes of GINs are excited, and their spontaneous activity is elevated, by tonic activation of nAChRs (*Luo et al., 2013*; *Muñoz-Manchado et al., 2014*, *Ibáñez-Sandoval*

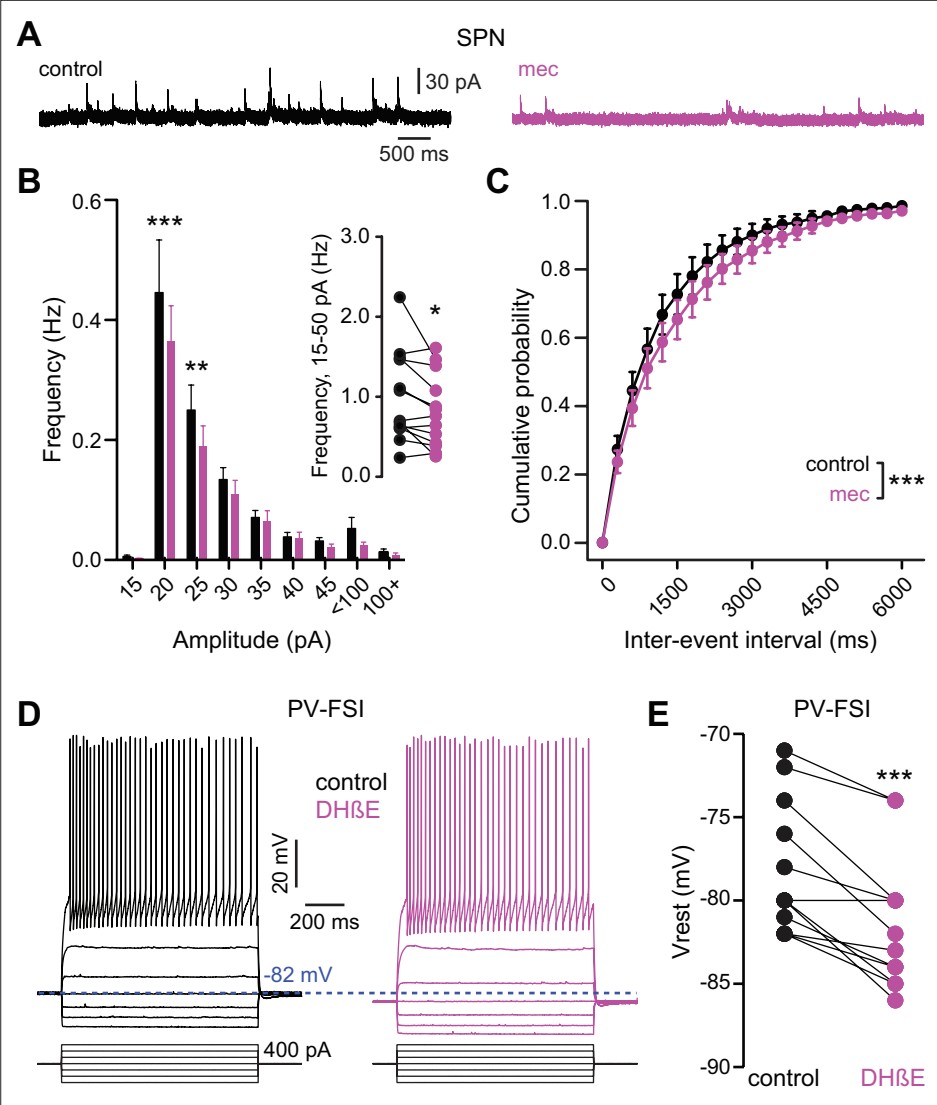

**Figure 7.** Nicotinic acetylcholine receptor (nAChR) blockade hyperpolarizes parvalbumin-positive fast-spiking interneurons (PV-FSI) resting membrane potential and reduces the frequency of spontaneous inhibitory postsynaptic currents (sIPSCs) in striatal spiny projection neurons (SPNs). (**A**) Example recordings of sIPSCs from SPNs voltage-clamped at +10 mV, before and during mecamylamine (10 μM) application. (**B**) Mecamylamine significantly enhanced sIPSC frequency in SPNs (F8,108=2.607, p=0.012; n=13; two-way ANOVA). A post hoc Bonferroni test revealed that this decrease was limited to low-amplitude sIPSCs (20 pA bin: p<0.001; 25 pA bin: p<0.01). (**C**) Mecamylamine caused a rightward shift in the cumulative probability distribution of sIPSC interevent intervals in SPNs (F(20,252)=2.10, p=4.7·10⁻³; n=13; two-way ANOVA). (**D**) Example traces of a PV-FSI before and after wash-in of 1 μM dihydro-β-erythroidine hydrobromide (DHβE). (**E**) Resting membrane potentials of PV-FSIs before and after wash-in of DHβE (n=12; p=9.8·10⁻⁴; signed-rank test [SRT]).

The online version of this article includes the following figure supplement(s) for figure 7:

**Figure supplement 1.** Nicotinic acetylcholine receptor (nAChR) blockade depolarizes somatostatin (SOM)+ interneuron membrane potential.

---

et al., 2015, **Elghaba et al., 2016**; **Tepper et al., 2018**). Because PV-FSIs express nAChRs and convey strong feedforward inhibition to SPNs (**Koós and Tepper, 1999**; **Tepper et al., 2004b**; **Planert et al., 2010**), but only weakly respond to phasic activation of CINs (**English et al., 2011**), we made targeted current-clamp recordings from PV-Cre × Ai9-tdTomato transgenic mice (**Johansson and Silberberg, 2020**) and tested the effect of nAChR blockade on their excitability (**Figure 7d**). Indeed, blocking nAChRs by bath application of DHβE significantly hyperpolarized the resting membrane potential of

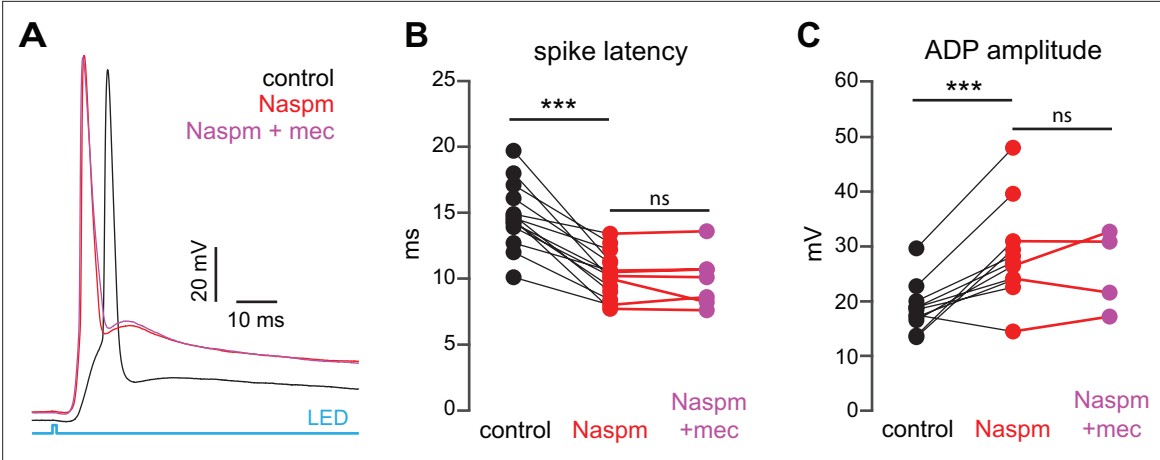

**Figure 8.** Blockade of Ca²⁺-permeable AMPA receptors, to prevent cortical activation of parvalbumin-positive fast-spiking interneurons (PV-FSIs) and related GABAergic interneurons (GINs), mimics and occludes the effect of mecamylamine on striatal spiny projection neuron (SPN) spike latency. (**A**) Example of the occlusion of the mecamylamine effect on optogenetic synaptic activation of SPNs by the Ca²⁺-permeable AMPA receptor antagonist 1-naphthyl acetyl spermine (Naspm) (100 µM). Distribution of spike latencies (**B**) and afterdepolarization (ADP) amplitude (**C**) in response to application of Naspm followed by mecamylamine show that Naspm significantly shortens the action potential (AP) latency (p=1.22·10⁻⁴, n=14 SPNs, signed-rank test [SRT]) and enhancement of the ADP amplitude (p=1.95·10⁻³, n=11 SPNs, SRT). In contrast, the subsequent mecamylamine application fails to further shorten the AP latency (p=0.6875, n=7 SPNs, SRT) or further enhance the ADP amplitude (p=0.625, n=4, SRT).

PV-FSIs (*Figure 7d–e*) – this effect of nAChR blockade was not universally observed in other populations of GINs that are involved in feedforward inhibition, such as spontaneously firing SOM+ interneurons, in which mecamylamine had no effect on firing rate and actually depolarized the average membrane potential (*Figure 7—figure supplement 1*). Thus, tonic activation of nAChRs depolarizes the resting membrane potential of PV-FSIs, continuously holding them closer to the AP threshold, and thereby priming them to transmit cortically driven feedforward inhibition more efficiently to SPNs. Interestingly, the dependence upon trailing 'primed' GIN input may explain why mecamylamine was capable of diminishing the amplitude of relatively slow rising EPSPs (*Figure 1e*) but not faster peaking excitatory postsynaptic currents (EPSCs) (*Figure 2b*).

If SPN spike latency is perpetually slowed due to tonic nAChR-mediated 'priming' of presynaptic GINs, then taking the relevant GINs offline should mimic and occlude the effect of mecamylamine. Unlike some other GINs that target SPNs, such as persistent/plateau-low-threshold spiking (LTS) interneurons (*Tepper et al., 2010*; *Plotkin and Goldberg, 2019*), synaptic responses of PV-FSIs to cortical inputs are mediated by Ca²⁺-permeable AMPA receptors that lack the GluA2 subunit (*Gittis et al., 2010*). In fact, pharmacological blockade of Ca²⁺-permeable GluA2-lacking AMPA receptor

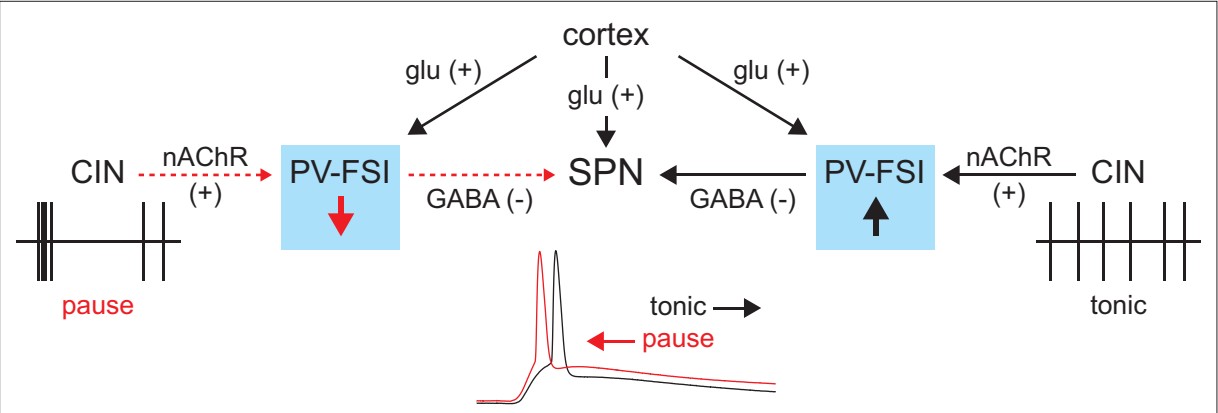

**Figure 9.** Proposed model of tonic nicotinic acetylcholine receptor (nAChR)-mediated suppression of cortically driven striatal spiny projection neuron (SPN) spike latency. Example cholinergic interneuron (CIN) and SPN firing patterns shown below; dashed red arrows indicate diminished signaling during CIN pauses; Glu = glutamate.

subunits prevents cortical activation of PV-FSIs but not persistent/plateau-LTS interneurons (*Gittis et al., 2010*; *Gittis et al., 2011*). Indeed, pharmacologically preventing cortical activation of PV-FSIs with the $Ca^{2+}$-permeable AMPA receptor antagonist 1-naphthyl acetyl spermine (Naspm; 100 µM) not only reduced spike latency and ADP amplitude, but occluded the effect of mecamylamine (*Figure 8*). Taken together, these data demonstrate that tonic activation of nAChRs on a subclass of GINs is responsible for delaying cortically evoked SPN spiking, and implicate PV-FSIs as the primary mediator.

In summary, the tonic activity of CINs recruits a subclass of GINs, namely PV-FSIs, to provide a 'nicotinic brake' on SPN responsiveness to cortical activity (*Figure 9*). Such a mechanism could allow SPNs to be more receptive of cortical inputs when CINs pause their firing in response to salient inputs or stimuli associated with reward.

## Discussion

The present study describes a novel role for tonic nAChR activation in shaping striatal output. Decades of research have firmly established CINs as important modulators – mostly via mAChRs (*Goldberg et al., 2012*; *Zucca et al., 2018*; *Assous, 2021*) – of cortico- and thalamo-striatal synaptic integration and determinants of striatal output (*Akins et al., 1990*; *English et al., 2011*; *Abudukeyoumu et al., 2019*). Despite the fact that CINs are autonomous pacemakers, nearly all studies examining their influence on the precise temporal output of the striatum, particularly via nAChRs, have been centered around the consequences of their phasic engagement (*Witten et al., 2010*; *English et al., 2011*; *Faust et al., 2016*; *Dorst et al., 2020*). Here, we report that ongoing, non-phasic activation of nAChRs accounts for a basal elevation in GABAergic tone, which dampens the response of SPNs to convergent cortical inputs and ultimately delays striatal output.

### Basal cholinergic regulation of striatal output – a postsynaptic 'nicotinic brake'?

CINs can robustly evoke GABAergic inhibition in the striatum by engaging somatodendritic nAChRs on specific classes of local GINs, in particular those expressing NPY (*English et al., 2011*; *Luo et al., 2013*). This cell-type-specific nAChR-dependent mechanism has been demonstrated using electrical or optogenetic stimulation to drive *phasic*, stimulation-locked inhibition. Somewhat puzzling, however, are reports that certain GINs, including PV-FSIs, only weakly respond to phasic activation of CINs or cholinergic afferents from the midbrain (*Dautan et al., 2020*) even though they express the requisite somatodendritic nAChRs (*Koós and Tepper, 2002*; *English et al., 2011*; *Nelson et al., 2014a*; *Nelson et al., 2014b*). Our finding that tonic activation of nAChRs enhances the excitability of FSIs, as has been shown previously (*Koós and Tepper, 2002*; *Luo et al., 2013*), offers a physiological role for these receptors. In fact, tonic activation of nAChRs on FSIs (driven by the tonic activity of CINs) may partly explain why previous studies have only observed modest responses of FSIs to additional phasic stimulation of CINs (*English et al., 2011*; *Nelson et al., 2014a*).

Tonic nAChR excitation of GINs helps clarify another puzzling observation. It is obvious why GABAR blockers occlude the effect of mecamylamine on AP latency and ADP amplitude in SPNs – the relevant GABARs are synaptically downstream of the nAChRs on GINs. But why is it that mecamylamine seems to 'saturate' the effect of GABAR inhibition on SPN AP latency and ADP? In other words, why does blockade of GABARs have no additional effect on AP latency or ADP when applied after mecamylamine, when phasic CIN activation only engages a select few types of GINs? If nAChRs were recruited solely by phasic activation of CINs, then the addition of GABAR blockers after mecamylamine should further shorten the AP latency and increase ADP amplitude by attenuating the influence of other GINs (such as PV-FSIs) that are only weakly responsive to phasic activation of nAChRs (*English et al., 2011*; *Nelson et al., 2014a*; *Nelson et al., 2014b*). A 'priming' effect of tonic nAChR activation on GINs that are not directly driven to spike by CINs could explain the seemingly oversized influence of nicotinic signaling on GABAergic inhibition. It is also worth noting that such 'priming' of feed forward inhibition (where SPNs and GINs are activated near-simultaneously by converging corticostriatal afferents) will ensure that this effect of GINs on SPNs is always inhibitory, despite the relatively depolarized reversal potential of $Cl^-$ in SPNs in vivo (*Bracci and Panzeri, 2006*), particularly when SPNs are at a near threshold membrane voltage (*Figure 1F,G*).

While an exhaustive categorization of the role every GIN class plays (or does not play) in mediating the observed effect may be ill-posed, we have identified PV-FSIs as the likely primary contributor. First, blockade of nAChRs hyperpolarized PV-FSIs at rest, suggesting that tonic activation of nAChRs holds these GINs closer to spike threshold. We observed the opposite effect in a similar class of GINs that mediates feedforward inhibition (SOM+ interneurons), making their involvement in the process less likely. While the effect on PV-FSI resting membrane potential that we observed was modest, we note that a similar magnitude mecamylamine-induced hyperpolarization is sufficient to slow the spontaneous firing rate of NPY-expressing LTS interneurons (*Elghaba et al., 2016*). Second, pharmacologically blocking cortical activation of PV-FSIs with a $Ca^{2+}$-permeable AMPA receptor antagonist, which leaves synaptic activation of other GINs such as persistent/plateau-LTS interneurons intact, completely occluded the effect of mecamylamine on SPN spike latency. Together, these data corroborate PV-FSIs as the primary candidate to exert the 'nicotinic brake' we observed during basal circuit activity.

## Under what conditions will the 'nicotinic brake' be lifted?

Through their continuous autonomous discharge, CINs tonically release ACh in the striatum (*Zhou et al., 2002*). While a prolonged exposure of striatal nAChRs to ACh is bound to desensitize them, it has been argued that the fast action of striatal acetylcholinesterase (AChE) prevents nAChR desensitization, allowing for the maintenance of tonic nAChR activation in the striatum (*Zhou et al., 2002*). So if tonic nAChR activation normally puts a brake on cortical feedforward inhibition, under what circumstances is this brake removed? The TANs of the primate striatum, which are comprised mainly of CINs (*Wilson et al., 1990*; *Aosaki et al., 1995*), acquire a synchronized pause response (*Raz et al., 1996*) in their tonic firing that lasts several hundred milliseconds in response to primary reward or, through conditioning, to salient stimuli associated with reward (*Kimura et al., 1984*; *Apicella et al., 1997*; *Goldberg and Reynolds, 2011*; *Bradfield et al., 2013*). This pause duration should be sufficient – particularly while AChE rapidly clears the extracellular space from ACh – to deactivate the nAChRs, thereby removing the nicotinic brake.

Our study complements a recent study that described how CINs impact SPN neural dynamics in vivo (*Zucca et al., 2018*). In that study it was shown that when CINs are optogenetically silenced in a synchronous fashion for a sufficiently long period, SPNs become hyperpolarized, less excitable, and short-term corticostriatal plasticity is dampened. However, these effects commence only 400 ms after CINs are silenced. Until that point SPN excitability is unaffected. Thus, that study remained agnostic about the effect of CIN pauses that are shorter than 400 ms long on SPNs. Because physiological TAN pauses in response to sensory cues are typically over after 300 ms (*Kimura et al., 1984*; *Apicella et al., 1997*), the mAChR-mediated curbing of SPN excitability and responsiveness revealed in that and other studies (*Ebihara et al., 2013*) may not be relevant to the role of the shorter, more physiological CIN pauses. Assuming that the nAChR-dependent mechanism we revealed can come into play sooner, our findings suggest that there are two dichotomous phases to the effect of synchronous CIN pauses: an 'early' period that is dominated by the consequences of removing tonic nAChR activation, and a more delayed period driven by diminished activation of mAChRs, where the responsiveness of SPNs to incoming cortical input is weakened, as shown previously (*Zucca et al., 2018*).

## Implications for tuning striatal circuit activity

Our data suggest that as CIN pauses develop ongoing GABAergic inhibition will be relieved. This could have the effect of sharpening the timing of cortically driven spiking by several milliseconds, ultimately shaping the moment-by-moment processing of striatal output. Because the TAN pause usually coincides with the arrival of a salient stimulus, it is precisely a moment when SPNs need to be more attuned to cortical input and respond more reliably. Given the relatively higher affinity of α4β2 vs. α7 nAChRs (*Albuquerque et al., 2009*), it is not surprising that we found α7-mediated boosting of presynaptic glutamate release to be insensitive to tonic ACh levels, setting the stage for a CIN pause-initiated drop in ACh levels to promote a temporal sharpening of any proceeding SPN spikes that are induced. Moreover, seminal experiments in primates have shown that CIN pauses become more prominently and reliably evoked by sensory stimuli over the course of Pavlovian conditioning (*Aosaki et al., 1994*; *Aosaki et al., 1995*). Together, this implies that removal of the 'nicotinic brake' may be a crucial component of striatal learning. While a several milliseconds shift in spike timing may not seem like much in the context of 100 s of milliseconds long CIN pauses or SPN state transitions,

it may be ample time to move a spike into or out of the time window for NMDA receptor-dependent coincidence detection to occur (*Hao and Oertner, 2012*).

Both dSPNs and iSPNs robustly support spike timing-dependent plasticity (*Pawlak and Kerr, 2008*; *Shen et al., 2008*). Under permissive conditions, constraining the relative timing of SPN APs to occur 5 ms after a corticostriatal synaptic burst efficiently potentiates the active synapses (*Shen et al., 2008*). In this view, decreasing the time to spike on the time scale we observe may be sufficient to bring the spike temporally close enough to NMDA receptor activation to engage the signaling cascades that are crucial for long-term potentiation. Such subtle nAChR-dependent tuning could be a way of 'distancing' those same cortical afferents from SPN activity under conditions of basal activity, allowing for the synapses encoding a particular sensorimotor cue to become potentiated only when deemed appropriate by a learned cholinergic signal. Expanding upon this possibility, it was recently shown that synapse-specific corticostriatal long-term potentiation requires the coordination of CIN pauses with DA release and SPN depolarization (*Reynolds et al., 2022*). It is tempting to speculate that a function of the CIN pause in the scenario is to create a window of plasticity by removing the 'nicotinic brake' to reduce synaptic inhibition and promote and adjust the latency of evoked spikes.

The extent to which tonic activation of nAChRs will be capable of influencing GABA release and SPN responses to cortical inputs will likely depend on the capacity of AChE activity to both allow for the detection of a CIN pause and minimize nAChR desensitization (*Zhou et al., 2002*). This brings up an important corollary of the 'nicotinic brake' hypothesis: the putative mechanism by which CIN pauses remove the nicotinic brake may be most robust in the AChE-enriched compartment of the striatum, also known as the matrix (as opposed to striosomes) (*Graybiel and Ragsdale, 1978*; *Brimblecombe and Cragg, 2017*; *Prager and Plotkin, 2019*). Because the matrix constitutes ~85% of the striatum, our data likely reflect the situation in that compartment (*Prager and Plotkin, 2019*). While coordinated CIN activation can interrupt the timing of ongoing APs in both matrix and striosome SPNs (*Crittenden et al., 2017*), the degree of tonic nAChR activation and existence of a 'nicotinic brake' in striosomes is unclear.

Taken together, our data suggest an additional role for cholinergic signaling in shaping striatal activity. While a great deal is known about how ACh can impact SPN function through metabotropic muscarinic receptors, nearly all studies of the 'faster' nAChR arm of ACh signaling have focused on its phasic activation. The data in this study demonstrate that nAChR signaling is not constrained to times of phasic CIN engagement, and tonic activation of nAChRs places a 'brake' on SPN activity that may help guide the fine tuning of striatal output.

# Materials and methods

**Key resources table**

| Reagent type (species) or resource | Designation | Source or reference | Identifiers | Additional information |
|---|---|---|---|---|
| Strain, strain background (*Mus musculus*) | C57BL/6J | Jackson Laboratory | RRID: IMSR_JAX:000664 | |
| Genetic reagent (*Mus musculus*) | B6.FVB-Tg(*Drd2*-EGFP/Rpl10a) CP101Htz/J | Jackson Laboratory | RRID: IMSR_JAX:030255 | |
| Genetic reagent (*Mus musculus*) | B6.Cg-Tg(*Drd1a*-tdTomato)6Calak/J | Jackson Laboratory | RRID: IMSR_JAX:016204 | |
| Genetic reagent (*Mus musculus*) | B6.Cg-Tg(*Thy1*-COP4/EYFP)18Gfng/J | Jackson Laboratory | RRID:IMSR_JAX:007612 | |
| Genetic reagent (*Mus musculus*) | *Pvalb*-cre | Jackson Laboratory | RRID: IMSR_JAX:017320 | |
| Genetic reagent (*Mus musculus*) | *Sst*-cre | Jackson Laboratory | RRID: IMSR_JAX:018973 | |
| Genetic reagent (*Mus musculus*) | Ai9 ('tdTomato') | Jackson Laboratory | RRID: IMSR_JAX:007909 | |
| Other | AAV5-Ef1a-DIO-EYFP | Addgene | 27056-AAV5 | *Adeno-associated virus* (AAV) |

*Continued on next page*

*Continued*

| Reagent type (species) or resource | Designation | Source or reference | Identifiers | Additional information |
|---|---|---|---|---|
| Chemical compound, drug | Mecamylamine hydrochloride | Sigma-Aldrich Tocris | Lot # 019M4108V CAS: 826-39-1 #2843, CAS: 110691-49-1 | |
| Chemical compound, drug | Dihydro-β-erythroidine hydrobromide | Tocris | #2349, CAS: 29734-68-7 | |
| Chemical compound, drug | SR 95531 hydrobromide (Gabazine) | Hello Bio | CAS: 104104-50-9 | |
| Chemical compound, drug | DNQX | TOCRIS | CAS: 2379-57-9 | |
| Chemical compound, drug | D-AP5 | Hello Bio | CAS:79055-68-8 | |
| Chemical compound, drug | Methyllycaconitine citrate | TOCRIS | Lot # 23A/255947 CAS:351344-10-0 | |
| Chemical compound, drug | Naspm trihydrochloride | Alomone Labs | CAS: 1049731-36-3 | |
| Chemical compound, drug | CGP 55845 hydrochloride | Hello Bio | CAS: 149184-22-5 | |
| Chemical compound, drug | XYLAZINE AS HYDROCHLORIDE | EUROVET ANIMAL HEALTH B.V | CAS: 082-91-92341-00 | |
| Chemical compound, drug | CLORKETAM | VETOQUINOL | CAS: 1867-66-9 | |
| Chemical compound, drug | VECTASHIELD Vibrance Antifade Mounting Medium with DAPI | VECTOR LABORATORIES | SKU: H-1800 | |
| Antibody | Recombinant Anti-GAD65+GAD67 antibody [EPR19366] | Abcam | ab183999 | Rabbit monoclonal (1:1000) |
| Antibody | Rabbit polyclonal anti Somatostatin-14 | Peninsula Laboratories | T-4102 | Rabbit polyclonal (1:100) |

## Animals

All experimental procedures on mice adhered to and received prior written approval from the Institutional Animal Care and Use Committees of the Hebrew University of Jerusalem (MD-14-14195-3 and MD-18-15657-3) and of Stony Brook University (737496) and of the local ethics committee of Stockholm, Stockholms Norra djurförsöksetiska nämnd (N2022_2020). Experiments were conducted on various transgenic mice. For optogenetic activation of corticostriatal fibers, we used 1- to 4-month-old male and female homozygous transgenic *Thy1*-ChR2 mice (B6.Cg-Tg(Thy1-COP4/EYFP)18Gfng/J) that express ChR2 under the *Thy1* promoter were used. *Thy1*-ChR2 mice express ChR2 in cortical afferents under the thymus cell antigen 1 (*Thy1*) promoter. *Thy1* is generally expressed in the axons of layer V pyramidal neurons, limbic system, midbrain, and brainstem (*Arenkiel et al., 2007*). For experiments measuring paired pulse ratios (PPRs), neuronal excitability and spontaneous GABAergic events, we used 1.5- to 4-month-old male and female C57BL/6 mice crossed with one of two BAC transgenic lines (where indicated): *drd1a*-tdTomato (labeling dSPNs) or *drd2*-eGFP (labeling iSPNs) (*Shuen et al., 2008*; *Ade et al., 2011*). For PV-FSI and SOM+ interneuron recordings, both male and female mice (postnatal days 47–82) were used in this study. Mice were group-housed under a 12 hr light/dark schedule and given ad libitum access to food and water. The *Pvalb*-Cre (stock #017320, the Jackson laboratory) mouse line was crossed with a homozygous tdTomato reporter mouse line ('Ai9', stock #007909, the Jackson laboratory) to allow identification of FSIs based on the expression of a fluorescent marker protein. SOM+ interneurons were identified by injecting cre-dependent AAV (27056-AAV5, Addgene) into the striatum of *Sst*-cre mice, inducing the expression of YFP in SOM-positive neurons specifically (stock #018973, the Jackson Laboratory). The PV- and SOM- Cre lines were heterozygous and maintained on a wild-type C57BL/6J background (stock # 000664, the Jackson Laboratory).

## Virus injections

SOM- cre mice were anesthetized with isoflurane and placed in a stereotaxic frame (Harvard Apparatus, Holliston, MA). A Quintessential Stereotaxic Injector (Stoelting, Wood Dale, IL) was used to

inject 0.5 µl of AAV5-EF1a-DIO-EYFP (#27056, Addgene) into the striatum (+0.5 AP, 2 ML, 2.5 DV) at a speed of 0.1 µl/min. The pipette was held in place for at least 5 min after the injection. Following the surgery, the mice were given analgesics (buprenorphine, 0.08 mg/kg, i.p.).

## Slice preparation

For optogenetic activation experiments, PPR, excitability and spontaneous GABAergic event experiments: Mice were deeply anesthetized with ketamine-xylazine and perfused transcardially with ice-cold modified artificial cerebrospinal fluid (ACSF) bubbled with 95% $O_2$–5% $CO_2$, and containing (in mM): 2.5 KCl, 26 $NaHCO_3$, 1.25 $Na_2HPO_4$, 0.5 $CaCl_2$, 10 $MgSO_4$, 0.4 ascorbic acid, 10 glucose, and 210 sucrose (optogenetic activation studies) or 3 KCl, 26 $NaHCO_3$, 1 $NaH_2PO_4$, 1 $CaCl_2$, 1.5 $MgCl_2$, 124 NaCl, and 14 glucose (PPR, excitability and spontaneous GABAergic event measures). The brain was removed and sagittal or coronal slices sectioned at a thickness of 275 µm were obtained in ice-cold modified ACSF. Slices were then submerged in ACSF, bubbled with 95% $O_2$–5% $CO_2$, and containing (in mM): 2.5 KCl, 126 NaCl, 26 $NaHCO_3$, 1.25 $Na_2HPO_4$, 2 $CaCl_2$, 2 $MgSO_4$, and 10 glucose and stored at room temperature for at least 1 hr prior to recording (optogenetic activation studies), or 3 KCl, 26 $NaHCO_3$, 1 $NaH_2PO_4$, 2 $CaCl_2$, 1 $MgCl_2$, 124 NaCl, and 14 glucose, incubated at 32°C for 45 min, then held at room temperature until recording (PPR, excitability and spontaneous GABAergic event measures).

For PV-FSI and SOM+ experiments: Mice were deeply anesthetized with isoflurane and decapitated. The brain was removed and immersed in ice-cold cutting solution containing 205 mM sucrose, 10 mM glucose, 25 mM $NaHCO_3$, 2.5 mM KCl, 1.25 mM $NaH2PO_4$, 0.5 mM $CaCl_2$, and 7.5 mM $MgCl_2$. Parasagittal brain slices (thickness 250 mm) were prepared with a Leica VT1000S vibratome and incubated for 30–60 min at 34°C in a submerged chamber filled with ACSF saturated with 95% oxygen and 5% carbon dioxide. ACSF was composed of 125 mM NaCl, 25 mM glucose, 25 mM $NaHCO_3$, 2.5 mM KCl, 2 mM $CaCl_2$, 1.25 mM $NaH_2PO_4$, 1 mM $MgCl_2$. Subsequently, slices were kept for at least 30 min at room temperature before recording.

## Slice visualization, electrophysiology, and optogenetic stimulation

For optogenetic activation experiments: The slices were transferred to the recording chamber mounted on an Olympus BX51 upright, fixed-stage microscope and perfused with oxygenated ACSF at room temperature. A 60×, 0.9 NA water immersion objective was used to examine the slice using Dodt contrast video microscopy. Patch pipette resistance was typically 3–4 MΩ when filled with recording solutions. In voltage-clamp experiments of IPSCs in SPNs, the intracellular solution contained (in mM): 127.5 $CsCH_3SO_3$, 7.5 CsCl, 10 HEPES, 10 TEA-Cl, 4 phosphocreatine disodium, 0.2 EGTA, 0.21 $Na_2GTP$, and 2 $Mg_{1.5}ATP$ (pH = 7.3 with CsOH, 280–290 mOsm/kg). For whole-cell current-clamp recordings from SPNs and cell-attached recordings from CINs, the pipette contained (in mM): 135.5 $KCH_3SO_4$, 5 KCl, 2.5 NaCl, 5 Na-phosphocreatine, 10 HEPES, 0.2 EGTA, 0.21 $Na_2GTP$, and 2 $Mg_{1.5}ATP$ (pH = 7.3 with KOH, 280–290 mOsm/kg). Electrophysiological recordings were obtained with a Multiclamp 700B amplifier (Molecular Devices, Sunnyvale, CA). Junction potential, which was 7–8 mV, was not corrected. Signals were digitized at 10 kHz and logged onto a personal computer with the Winfluor software (John Dempster, University of Strathclyde, UK). Blue light LED (470 nm, Mightex, Toronto, Ontario, Canada) was used for full-field illumination via the objective. Single pulses were 1 ms long. In order to compare AP latency and ADP amplitude among SPNs, the LED intensity was set to generate a just suprathreshold response in each SPN recorded.

For PPR, excitability and spontaneous GABAergic event experiments: The slices were transferred to the recording chamber mounted on a modified Ultima laser scanning microscope system (Bruker Nano) (excitability and spontaneous GABAergic event experiments) or an Olympus BX51W1 microscope (PPR experiments) and perfused with oxygenated ACSF at room temperature. A 60×, 1.0 NA Olympus LUMPFL water immersion objective was used to visualize slices, and fluorescently labeled neurons identified for recordings with the aid of a Dodt contrast image displayed in registration with the fluorescence image for experiments requiring identification of dSPNs vs. iSPNs. Patch pipette resistance was typically 3–6 MΩ when filled with recording solutions. For voltage-clamp recordings, the intracellular solution contained (in mM): 120 $CsMeSO_3$, 5 NaCl, 10 TEA-Cl (tetraethylammonium-Cl), 10 HEPES, 5 Qx-314, 4 $ATP-Mg^{2+}$, 0.3 $GTP-Na^+$, 0.25 EGTA, and 0.05 Alexa Fluor 568 hydrazide $Na^+$salt (Alexa Fluor was omitted in PPR experiments). For current-clamp recordings, the

internal solution contained (in mM): 135 KMeSO$_4$, 5 KCl, 10 HEPES, 2 ATP-Mg$^{2+}$, 0.5 GTP-Na$^+$, 5 phosphocreatine-tris, 5 phosphocreatine-Na$^+$, 0.1 Fluo-4 pentapotassium salt, and 0.05 Alexa Fluor 568 hydrazide Na$^+$ salt. Patched SPNs were allowed to equilibrate for 10 min after rupture. Recordings were made using a Multiclamp 700B amplifier and either PrairieView 5.0 software (Bruker) or custom MATLAB protocols. sIPSCs were recorded in voltage-clamp mode at +10 mV for 180 s before and after bath application of drug. Access resistance and holding current were monitored throughout experiments and cells were excluded if these values changed by more than 20%. Local electrical stimulation was evoked using a concentric bipolar stimulation electrode (FHC, Inc, Bowdoin, ME) placed near the cell using a stimulating amplitude of ~1–3 mV and 50 ms between stimuli pulses; 10 sweeps (10 s intersweep interval) were averaged per cell; cells were held at –70 mV. PPR experiments were performed at 26–29°C. Rheobase and AP firing frequencies were determined in current-clamp mode via a step series of 500 ms current injections starting at –200 pA and increasing by 25 pA with each step. Resting membrane potential and cell morphology were monitored throughout experiments and cells were excluded if resting at a more depolarized potential than –77.0 mV.

For PV-FSI and SOM+ experiments: Whole-cell patch clamp recordings were obtained in oxygenated ACSF at 35°C. Neurons were visualized using infrared differential interference contrast microscopy (Zeiss FS Axioskop, Oberkochen, Germany). tdTomato- or EYFP-expressing neurons were identified by switching to epifluorescence using a mercury lamp (X-cite, 120Q, Lumen Dynamics). Borosilicate pipettes of 5–8 MOhm resistance were pulled with a Flaming/Brown micropipette puller P-1000 (Sutter instruments). All recordings were done in current clamp with an intracellular solution containing 130 mM K-gluconate, 5 mM KCl, 10 mM HEPES, 4 mM Mg-ATP, 0.3 mM GTP, 10 mM Na$_2$-phospho-creatine (pH 7.25, osmolarity 285 mOsm). Recordings were amplified using a MultiClamp 700B amplifier (Molecular Devices, San Jose, CA), filtered at 2 kHz, digitized at 10–20 kHz using ITC-18 (HEKA Elektronik GmbH, Germany), and acquired using custom-made routines running on Igor Pro (Wavemetrics, Portland, OR). Liquid junction potential was not corrected. Throughout all recordings, pipette capacitance and access resistance were compensated for and data were discarded when access resistance increased beyond 30 MOhm. The intrinsic properties of the neurons were determined by a series of hyperpolarizing and depolarizing current steps and ramps, enabling the extraction of sub- and suprathreshold properties. All neurons were recorded in control conditions and after >5 min of bath application of 1 μM DHβE (Tocris) or 10 μM Mec (Tocris).

## Drugs and reagents

Experiments in Thy1-ChR2 mice were performed in the presence or absence of synaptic receptor blockers including 10 μM SR-95531 (gabazine) to block GABA$_A$ receptors, 2 μM CGP-55845 to block GABA$_B$ receptors, 10 μM mecamylamine, a non-selective nAChR-antagonist, 1 or 10 μM DHβE, a competitive α4β2 nAChR antagonist and Naspm (100 μM), a Ca$^{2+}$-permeable AMPA receptor antagonist. Experiments were conducted with various combinations of the blockers. The acute effects of solution exchanges were measured at least 5 min after wash on. All drugs and reagents were acquired from Tocris (Ellisville, MO), Sigma (St Louis, MO) or HelloBio (Bristol, Avonmouth, UK).

## Histology

Nine-week-old female mice were deeply anesthetized with ketamine-xylazine followed by cold perfusion to the heart of PBS and 4% PFA. The removed brain was kept overnight at 4°C in 4%–4% PFA. The next day, the brain was washed in PBS (three times, 20 min each) before thin 40 μm coronal slices were made using a vibratome (Leica VT1000S). Slices were briefly washed with CAS-BLOCK (Life Technologies) before being incubated in CAS-BLOCK (300 μl) overnight with GAD65+67 antibodies (ab183999 Abcam, 1:1000) or Somatostatin (BMA Biomedicals Peninsula Laboratories T-4102, 1:100). The slices were then washed in PBS (three times, 20 min each) and incubated with CAS-BLOCK and secondary antibodies (Abcam ab150063, 1:500) for 3 hr. Antifade Mounting Medium (VECTASHIELD) was applied to prevent slice bleaching.

## Data analysis and statistics

Data analysis was performed using custom-made code in MATLAB (MathWorks, Natick, MA) or Python; sIPSCs were analyzed using the Mini Analysis program (Synaptosoft). Two-tailed Wilcoxon SRT was used to test for changes of medians in matched-paired comparisons. The null hypotheses were

rejected if the p values were below 0.05. Boxplots represent range (whiskers), median (thick bar), and lower and upper quartiles. To analyze responses to stimulation, peristimulus time histograms (PSTHs) were generated. PSTHs were estimated using 20 ms wide bins. IV relationships, sIPSC frequencies, and interevent intervals were analyzed by two-way ANOVA.

## Additional information

### Funding

| Funder | Grant reference number | Author |
|---|---|---|
| BSF | 2017020 | Joshua A Goldberg Joshua L Plotkin |
| Israel Science Foundation | 154/14 | Joshua A Goldberg |
| European Research Council | Consolidator grant 646880 | Joshua A Goldberg |
| National Institute of Neurological Disorders and Stroke | R01 NS104089/NINDS | Joshua L Plotkin |
| National Institute of Neurological Disorders and Stroke | NS022061/NINDS | Joshua L Plotkin |
| Hjärnfonden | FO2021-0333 | Gilad Silberberg |
| Hjärnfonden | PS2020-0020 | Yvonne Johansson |
| Vetenskapsrådet | 2019-01254 | Gilad Silberberg |
| Wallenberg Academy Fellowship | KAW 2017.0273 | Gilad Silberberg |

The funders had no role in study design, data collection and interpretation, or the decision to submit the work for publication.

### Author contributions

Lior Matityahu, Jeffrey M Malgady, Meital Schirelman, Yvonne Johansson, Jennifer A Wilking, Formal analysis, Investigation; Gilad Silberberg, Joshua A Goldberg, Joshua L Plotkin, Conceptualization, Funding acquisition, Project administration, Supervision, Writing - original draft, Writing - review and editing

### Author ORCIDs

Lior Matityahu http://orcid.org/0000-0002-6115-8608
Jeffrey M Malgady http://orcid.org/0000-0002-1129-2155
Meital Schirelman http://orcid.org/0000-0002-9081-7019
Yvonne Johansson http://orcid.org/0000-0001-9781-9204
Jennifer A Wilking http://orcid.org/0000-0002-6488-412X
Gilad Silberberg http://orcid.org/0000-0001-9964-505X
Joshua A Goldberg http://orcid.org/0000-0002-5740-4087
Joshua L Plotkin http://orcid.org/0000-0001-6232-7613

### Ethics

All experimental procedures on mice adhered to and received prior written approval from the Institutional Animal Care and Use Committees of the Hebrew University of Jerusalem (MD-14-14195-3 and MD-18-15657-3) and of Stony Brook University (737496) and of the local ethics committee of Stockholm, Stockholms Norra djurförsöksetiska nämnd (N2022_2020).

### Decision letter and Author response

Decision letter https://doi.org/10.7554/eLife.75829.sa1
Author response https://doi.org/10.7554/eLife.75829.sa2

## Additional files

### Supplementary files
• Transparent reporting form

### Data availability
All analyzed data sets, whether included in figures or referenced as 'not shown', have been uploaded to OSF and made publically available: https://osf.io/7kazd.

The following dataset was generated:

| Author(s) | Year | Dataset title | Dataset URL | Database and Identifier |
|---|---|---|---|---|
| Plotkin JL, Goldberg J, Silberberg G | 2022 | Matityahu et al | https://osf.io/7kazd | Open Science Framework, 7kazd |

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
