## [Editor Report]

Matityahu et al., investigate the influence of nicotinic acetylcholine receptor signaling on striatal microcircuit function through a combination of slice electrophysiology, optogenetics, and pharmacology. They find that nicotinic signaling delays spiking of striatal projection neurons in response to excitatory input, likely through the tonic release of acetylcholine by cholinergic interneurons onto local GABAergic interneurons and their influence on striatal projection neurons. Understanding how acetylcholine shapes striatal circuits is important, as this neurotransmitter is implicated in multiple movement disorders as well as other basal ganglia-related diseases.

---

## [Decision Letter]

**Decision letter after peer review:**

Thank you for submitting your article "Tonic engagement of nicotinic receptors bidirectionally controls striatal spiny projection neuron spike timing" for consideration by *eLife*. Your article has been reviewed by 3 peer reviewers, and the evaluation has been overseen by a Reviewing Editor and John Huguenard as the Senior Editor. The following individual involved in review of your submission has agreed to reveal their identity: Fu-Ming Zhou (Reviewer #1).

Essential revisions:

1) Please provide sufficient and specific interpretation of the existing data. Some additional experiments are needed to make the conclusion more convincing, for example, increase n number, clearer definition of ADP. Please see Reviewer 1's detailed suggestions.

2) Reviewer 3 expressed concerns about using the Thy1-ChR2 line, and suggested using an additional cortical specific mouse line to strengthen the conclusion. The authors can either follow this suggestion to repeat some experiments using a new mouse line, or the authors can provide evidence showing expression of Thy1-ChR2 is restricted in cortical excitatory neurons (histology) and measure the latencies for excitatory vs inhibitory responses to verify there are no monosynaptic inhibitory components.

3) The authors highlight a mechanism by which cortical excitation would engage cholinergic and GABAergic neurons ending in GIN-mediated delay of spiking in SPNs. But they do relatively little to identify which GABAergic cell types are likely to contribute. They look specifically at how DHBE increases the excitability of PV-FSI, but provide neither direct evidence for PV-FSI nor rule our other cell types. Additional experiments are essential to address this major concern.

4) Please conduct PPR experiments for MLA and add the results in Figure 2.

5) Please provide data where it says "data not shown".

6) Reviewers provided detailed suggestions, and all reviewers agree that these are reasonable, please edit accordingly.

*Reviewer #1 (Recommendations for the authors):*

"Engagement" is used many times in the text. But this word is not informative. "engagement of nicotinic receptors" (used in the title) is not even logical: a receptor is either activated or inactivated or inhibited. "Spike advancement" is similarly problematic.

I suggest that the authors make a summary diagram, helping the reader understand their new findings.

*Reviewer #2 (Recommendations for the authors):*

1. The authors state "To interrogate microcircuit-level responses of SPNs to cortical excitation of the striatum, we generated acute ex vivo brain slices from Thy1-ChR2 mice." The Thy1-ChR2 mouse line was originally billed as expressing ChR2 in excitatory cortical projection neurons, but this expression is not limited to cortical projection neurons. Even modest expression of ChR2 in other types of neurons, especially GABAergic neurons, could influence the results the authors show in Figures 1, 2, and 3, since they are using terminal field stimulation in the striatum. It also could influence the interpretation of the microcircuits engaged. Though not every experiment in the paper relies on this line, several do, so it would be helpful to (a) show by histology where expression is and is not present, perhaps even looking for GABAergic neurons amongst those ChR2-expressing neurons in the cortex in the line, (b) determine the timing of any GABAergic versus expected glutamatergic components of monosynaptic inputs to the striatum in the Thy1-ChR2 line, for example by measuring the latency to the first deflection from the onset of an optical stimulus, and (c) validate that the same change in spike delay can be seen with an approach targeted specifically to excitatory cortical projection neurons, such as using a Cre recombinase line selective for these neurons and a focal injection of Cre-dependent ChR2 in one of the cortical areas that projects to their recording site.

2. The authors' logic is not watertight in a paragraph about potential sources of IPSCs to SPNs and their modulation. SPNs receive inhibitory input from (at minimum) other SPNs, fast-spiking interneurons, various types of other GABAergic interneurons, and dopamine neurons, as the authors note. They then state "SPNs are quiescent in acute slices and lack nAChRs, the source of these attenuated sIPSCs is likely GINs." However, sIPSCs are often (including in the striatum) a mixture of mIPSCs (spontaneous fusion events, not action potential-dependent) and action potential-dependent release. Though SPNs and fast-spiking interneurons are typically highly hyperpolarized in the slice preparation, they do contribute to mIPSCs and to the total sIPSCs recorded. The action potential-dependent sources are more likely to be from spontaneously firing GINs, such as the low-threshold-spiking types, or from axo-axonic/disynaptic release events driven by spontaneously active populations. In the same section, the authors state "Because PV-FSIs convey strong feedforward inhibition to SPNs (Koós and Tepper 1999, Tepper, Koós et al., 2004, Planert, Szydlowski et al., 2010), but do not respond to phasic CIN (English, Ibanez-Sandoval et al., 2011)"…This point is brought up again in the Discussion. However, in the cited paper, English et al., do not say that PV-FSI do not respond to optical stimulation of CIN, merely that they don't have large responses (in that paper they say there are no responses measured in current clamp of >3 mV). In another paper, measurements in voltage-clamp also showed modest nicotinic responses in PV-FSI of about 20 pA (Nelson, Bussert et al., 2014), also indicating such responses are small, but not nonexistent. Given the authors' findings, in fact the high tonic activity of CINs in slice preparations at physiological temperatures, that may be why English et al., and Nelson et al., found relatively modest responses to phasic activation. These points do not take away at all from the authors findings, but the language in the Results and Discussion could be made more precise/accurate.

3. The authors highlight a mechanism by which (presumed) cortical excitation would engage cholinergic and GABAergic neurons ending in GIN-mediated delay of spiking in SPNs. But they do relatively little to identify which GABAergic cell types are likely to contribute. They look specifically at how DHBE increases excitability of PV-FSI, which is suggestive, but provide neither direct evidence for PV-FSI nor rule our other cell types. The role of specific GABAergic interneurons could be shown more directly by manipulations of PV-FSI (or other cell types) within their microcircuit activation protocol. These could include PV-specific chemogenetic/optogenetic manipulations, or Cre-dependent cell killing methods, or possibly use of drugs that specifically inhibit Ca-permeable AMPA receptors, such as are found on PV-FSI in the striatum (though I am not sure if these types of AMPA receptors might also be expressed on CIN) (Gittis et al., J. Neurosci 2010; Gittis et al., J. Neurosci 2011).

*Reviewer #3 (Recommendations for the authors):*

1) Please add DhβE data in Figure 1 instead of "data not shown".

2) Please conduct PPR experiments for MLA and add the results in Figure 2.

3) Figure 5. Indicates "phasic activation of CINs cannot explain the nAChR-dependent delay of spike latency". This is not clear or straightforward to the reviewer. Please describe and explain more clearly in the text.

4) Please add Witten et al., 2010, PMID: 21164015 in the discussion.

---

## [Author Response]

Essential revisions:1) Please provide sufficient and specific interpretation of the existing data. Some additional experiments are needed to make the conclusion more convincing, for example, increase n number, clearer definition of ADP. Please see Reviewer 1's detailed suggestions.

We have addressed all of these comments in our revised manuscript, which we believe is now significantly stronger. Please see our point-by-point responses to Reviewer 1 below.

2) Reviewer 3 expressed concerns about using the Thy1-ChR2 line, and suggested using an additional cortical specific mouse line to strengthen the conclusion. The authors can either follow this suggestion to repeat some experiments using a new mouse line, or the authors can provide evidence showing expression of Thy1-ChR2 is restricted in cortical excitatory neurons (histology) and measure the latencies for excitatory vs inhibitory responses to verify there are no monosynaptic inhibitory components.

We appreciate this concern, and have addressed it experimentally with several new sets of data. We have added immunohistochemical data demonstrating that we see no evidence for ChR2 expression in GABAergic neurons in the cortex. That said, we did observe ChR2 in a population of pallidal neurons, as well as a detectible monosynaptic IPSC in SPNs in response to optogenetic stimulation. We present new data showing that this direct GABAergic input, however, is 1) non-cortical and extrastriatal in origin, 2) insensitive to nAChR blockade and 3) does not contribute to role of tonic nAChR activation in delaying cortically-driven SPN spike latency or ADPs. Please see our point-by-point responses to the Reviewers below.

3) The authors highlight a mechanism by which cortical excitation would engage cholinergic and GABAergic neurons ending in GIN-mediated delay of spiking in SPNs. But they do relatively little to identify which GABAergic cell types are likely to contribute. They look specifically at how DHBE increases the excitability of PV-FSI, but provide neither direct evidence for PV-FSI nor rule our other cell types. Additional experiments are essential to address this major concern.

We agree that our initial submitted manuscript made minimal attempt to identify the type of GINs responsible for mediating the effect of tonic nAChR activation on SPN spike latency. We have added 2 sets of new data to rectify this. First, we provide new data showing that blocking tonically activated nAChRs affects two different classes of GINs (PV-FSI and SOM+ interneurons) in different and opposing ways (with the impact on SOM+ interneurons not consistent with a role for them mediating our observed effect on SPN spike latency). Second, we added a new experiment – which was aptly suggested by Reviewer 2 – where cortical activation of PV-FSIs (but not other GIN types such as PLTS interneurons) was pharmacologically prevented using a blocker of GluA2-lacking AMPA receptors. Blocking synaptic activation of PV-FSIs completely mimicked and occluded the effect of mecamylamine on SPN spike latency. Please see our point-by-point responses to the Reviewers below.

4) Please conduct PPR experiments for MLA and add the results in Figure 2.

We have added this data.

5) Please provide data where it says "data not shown".

We have added this data.

6) Reviewers provided detailed suggestions, and all reviewers agree that these are reasonable, please edit accordingly.

We thank the Editor and Reviewers for their suggestions and efforts. We have edited accordingly, and believe our revised manuscript is significantly stronger.

Reviewer #1 (Recommendations for the authors):"Engagement" is used many times in the text. But this word is not informative. "engagement of nicotinic receptors" (used in the title) is not even logical: a receptor is either activated or inactivated or inhibited. "Spike advancement" is similarly problematic.

We have corrected our useage of “engagement” and “spike advancement” throughout the manuscript. The title of the manuscript was altered as well.

I suggest that the authors make a summary diagram, helping the reader understand their new findings.

Excellent suggestion. We have added a summary diagram, which is now figure 9.

Reviewer #2 (Recommendations for the authors):1. The authors state "To interrogate microcircuit-level responses of SPNs to cortical excitation of the striatum, we generated acute ex vivo brain slices from Thy1-ChR2 mice." The Thy1-ChR2 mouse line was originally billed as expressing ChR2 in excitatory cortical projection neurons, but this expression is not limited to cortical projection neurons. Even modest expression of ChR2 in other types of neurons, especially GABAergic neurons, could influence the results the authors show in Figures 1, 2, and 3, since they are using terminal field stimulation in the striatum. It also could influence the interpretation of the microcircuits engaged. Though not every experiment in the paper relies on this line, several do, so it would be helpful to (a) show by histology where expression is and is not present, perhaps even looking for GABAergic neurons amongst those ChR2-expressing neurons in the cortex in the line, (b) determine the timing of any GABAergic versus expected glutamatergic components of monosynaptic inputs to the striatum in the Thy1-ChR2 line, for example by measuring the latency to the first deflection from the onset of an optical stimulus, and (c) validate that the same change in spike delay can be seen with an approach targeted specifically to excitatory cortical projection neurons, such as using a Cre recombinase line selective for these neurons and a focal injection of Cre-dependent ChR2 in one of the cortical areas that projects to their recording site.

Excellent points. We have added new data to address these concerns (please see our response to the Public Reviewer Comments; data is presented in figure 3 supplement and updated text on pages 7-8). Briefly, we added new histological data showing that we found no evidence for ChR2 expression in GABAergic neurons within the cortex, though we did observe some non-cortical and extrastriatal expression. New experiments discovered that, as the Reviewer implied may be the case, there is indeed a monosynaptic GABAergic current that is evoked in SPNs by our optogenetic stimulation protocol. We confirmed the monosynaptic nature of this input by pharmacologically blocking AMPA and NMDA receptors. Importantly, however, this GABAergic input is not sensitive to mecamylamine and so it is does not impact our results and interpretation of the microcircuits involved in the phenomenon we describe. Given the non-cortical and mecamylamine-insensitive nature of this direct GABAergic input, weighed against the cost and long delay it would take to import a cre-dependent cortical mouse line (and limited additional insight it would provide given our new data) we feel that the newly added data addresses the spirit of the Reviewer’s main concerns.

2. The authors' logic is not watertight in a paragraph about potential sources of IPSCs to SPNs and their modulation. SPNs receive inhibitory input from (at minimum) other SPNs, fast-spiking interneurons, various types of other GABAergic interneurons, and dopamine neurons, as the authors note. They then state "SPNs are quiescent in acute slices and lack nAChRs, the source of these attenuated sIPSCs is likely GINs." However, sIPSCs are often (including in the striatum) a mixture of mIPSCs (spontaneous fusion events, not action potential-dependent) and action potential-dependent release. Though SPNs and fast-spiking interneurons are typically highly hyperpolarized in the slice preparation, they do contribute to mIPSCs and to the total sIPSCs recorded. The action potential-dependent sources are more likely to be from spontaneously firing GINs, such as the low-threshold-spiking types, or from axo-axonic/disynaptic release events driven by spontaneously active populations. In the same section, the authors state "Because PV-FSIs convey strong feedforward inhibition to SPNs (Koós and Tepper 1999, Tepper, Koós et al., 2004, Planert, Szydlowski et al., 2010), but do not respond to phasic CIN (English, Ibanez-Sandoval et al., 2011)"…This point is brought up again in the Discussion. However, in the cited paper, English et al., do not say that PV-FSI do not respond to optical stimulation of CIN, merely that they don’t have large responses (in that paper they say there are no responses measured in current clamp of >3 mV). In another paper, measurements in voltage-clamp also showed modest nicotinic responses in PV-FSI of about 20 pA (Nelson, Bussert et al., 2014), also indicating such responses are small, but not nonexistent. Given the authors’ findings, in fact the high tonic activity of CINs in slice preparations at physiological temperatures, that may be why English et al., and Nelson et al., found relatively modest responses to phasic activation. These points do not take away at all from the authors findings, but the language in the Results and Discussion could be made more precise/accurate.

These are excellent and very insightful points. We have removed our implication that the hyperpolarized nature of SPNs may preclude them from contributing to the measured sIPSCs (which we agree the Reviewer is correct about) and, to exercise caution, remain agnostic about the class of GINs contributing to the iPSCs we observe. Thank you for correcting us on our description of transient nAChR-mediated responses in PV-FSIs. We have updated the text throughout to clarify that this class of GINs does indeed show responses to transient activation of nAChRs, albeit weak/modest. The Reviewer’s suggestion that tonic activation of nAChRs on PV-FSIs in slice recordings may explain why other groups have only found relatively modest responses to phasic activation of nAChRs is an excellent one, and we have added this (with references) to the text (second paragraph of the discussion).

3. The authors highlight a mechanism by which (presumed) cortical excitation would engage cholinergic and GABAergic neurons ending in GIN-mediated delay of spiking in SPNs. But they do relatively little to identify which GABAergic cell types are likely to contribute. They look specifically at how DHBE increases excitability of PV-FSI, which is suggestive, but provide neither direct evidence for PV-FSI nor rule our other cell types. The role of specific GABAergic interneurons could be shown more directly by manipulations of PV-FSI (or other cell types) within their microcircuit activation protocol. These could include PV-specific chemogenetic/optogenetic manipulations, or Cre-dependent cell killing methods, or possibly use of drugs that specifically inhibit Ca-permeable AMPA receptors, such as are found on PV-FSI in the striatum (though I am not sure if these types of AMPA receptors might also be expressed on CIN) (Gittis et al., J. Neurosci 2010; Gittis et al., J. Neurosci 2011).

As pointed out by the Reviewer, our initial attempt to pin the described mechanism on a particular class of GINs was minimal at best. We thank the reviewer for the outstanding suggestion of targeting PV-FSIs pharmacologically by blocking GluA2-lacking AMPA receptors (side note: because the phenomenon we examined does not require and occurs before the consequences of phasic CIN activation, the presence or absence of these AMPA receptors on CINs should have little consequence on our interpretation). We have added new data (figure 8) showing that blocking cortical activation of PV-FSIs both mimicked and occluded the effect of mecamylamine on spike latency, implicating this class of GINs as a primary mediator of the effect and excluding others (such as PLTS interneurons) as participants.

Reviewer #3 (Recommendations for the authors):1) Please add DhβE data in Figure 1 instead of "data not shown".

We have added this data, now shown in the supplement to figure 1.

2) Please conduct PPR experiments for MLA and add the results in Figure 2.

We have added this data, now shown in the supplement to figure 2.

3) Figure 5. Indicates "phasic activation of CINs cannot explain the nAChR-dependent delay of spike latency". This is not clear or straightforward to the reviewer. Please describe and explain more clearly in the text.

We apologize for the confusion. Our reasoning was that the CIN-GIN-SPN disynaptic signal is too slow to mediate a change in spike latency on the time scale we report. We have clarified this text.

4) Please add Witten et al., 2010, PMID: 21164015 in the discussion.

We have added this reference (first paragraph of the discussion).